# Patient-Derived Xenograft Models in Urological Malignancies: Urothelial Cell Carcinoma and Renal Cell Carcinoma

**DOI:** 10.3390/cancers12020439

**Published:** 2020-02-13

**Authors:** Andrew T. Tracey, Katie S. Murray, Jonathan A. Coleman, Kwanghee Kim

**Affiliations:** 1Urology Service, Department of Surgery, Memorial Sloan Kettering Cancer Center, New York, NY 10065, USA; traceya@mskcc.org (A.T.T.); colemanj@mskcc.org (J.A.C.); 2Department of Surgery, Division of Urology, University of Missouri, Columbia, MO 65211, USA; murrayks@gmail.com; 3Department of Surgery, Memorial Sloan Kettering Cancer Center, New York, NY 10065, USA

**Keywords:** xenograft, renal cell carcinoma, urothelial cell carcinoma, patient-derived xenograft, urologic oncology

## Abstract

The engraftment of human tumor tissues into immunodeficient host mice to generate patient-derived xenograft (PDX) models has become increasingly utilized for many types of cancers. By capturing the unique genomic and molecular properties of the parental tumor, PDX models enable analysis of patient-specific clinical responses. PDX models are an important platform to address the contribution of inter-tumoral heterogeneity to therapeutic sensitivity, tumor evolution, and the mechanisms of treatment resistance. With the increasingly important role played by targeted therapies in urological malignancies, the establishment of representative PDX models can contribute to improved facilitation and adoption of precision medicine. In this review of the evolving role of the PDX in urothelial cancer and kidney cancer, we discuss the essential elements of successful graft development, effective translational application, and future directions for clinical models.

## 1. Introduction

The increasingly recognized complexity of human cancers and the heterogeneity of malignant tumor cells have posed a major challenge to development of effective therapies in urologic oncology. Traditional cancer cell lines, the backbone of pre-clinical oncologic research, often fail to adequately address many of these conundrums, lacking the ability to faithfully recapitulate cancer treatment response and malignant progression. The engraftment of patient solid tumor tissues into immunodeficient host mice to generate patient-derived xenograft (PDX) models has been shown to provide a more valuable predictive paradigm in today’s era of personalized medicine and targeted therapies [1,2]. In principle, such PDX models should capture the unique genomic and molecular properties of the parental patient tumor and enable analysis of patient-specific clinical responses to therapy [3].

The advent of PDX techniques date back to 1969, when a human colon cancer tumor was grafted into an athymic nude mouse [4]. However, the first reports of utilization of this technique on urologic malignancies, namely urothelial carcinoma and renal cell carcinoma (RCC), were published in the 1970′s [5,6,7]. Since this time, PDX models have increased in their prevalence, serving to improve our understanding of both bladder and kidney cancers.

In what has been termed the age of “precision medicine,” there is increasingly recognized heterogeneity between tumors that contributes to variation in therapeutic responses. PDX models can play a role in assessing susceptibility to both traditional chemotherapeutic agents, as well as novel targeted treatments that exploit the biology of a patient’s unique tumor type. Creation of a xenograft model involves transfer of tumor fragments from a live patient’s primary tumor or metastatic site into immune compromised mouse strains, most frequently NOD (non-obese diabetic) scid (severe combined immunodeficient) or NOD.Cg-Prkdc^scid^ Il2rg^tm1Wjl^/SzJ strains (NOD scid gamma, NSG). When utilized for research purposes, many studies have shown that these tumorgrafts retain the parental tumor histology, cellular structure, and genomic characteristics. At present, the cost of this process, variability in graft success rates, and the time required to establish the xenograft have been a barrier to the widespread implementation in clinical practice. While the oncology community remains optimistic that PDX models will one day serve as “mouse avatars,” providing patients and their clinicians with real-time information on tumor therapy response and tumor evolution, PDX models are not yet capable of serving as a decision-making tool. However, their role in the study of kidney and bladder cancers continues to expand, improving our understanding of disease biology.

Kidney cancer and bladder cancer have a major impact on the American public, together accounting for more than 150,000 new cases annually, and comprising 8.8% of new cancer diagnoses in 2019. Moreover, there will be a total of 33,420 patients who succumb to these diseases, accounting for 5.5% of all cancer-related deaths [8]. Though urothelial cell carcinoma and renal cell carcinoma are morphologically and histologically distinct malignancies, there are overlapping treatment paradigms in the advanced setting that make them both ideal tumors to study using PDX models. This is particularly true with the advent of next-generation sequencing and the revolution of tumor characterization in the “-omics” era.

Urothelial carcinoma, which originates in the lining of the bladder and upper tract, is the sixth most common malignancy in the United States. However, it has an outsized impact on the US healthcare system, with the highest lifetime treatment cost per case of all cancers [9]. Though effective endoscopic management and intravesical therapies portend a relatively good prognosis for non-muscle invasive bladder cancer, the survival trends for muscle-invasive and metastatic disease have stagnated for the past 30 years despite increasing utilization of neoadjuvant chemotherapy and advancements in surgical technique [10,11]. Though there have been encouraging results from clinical trials involving newly developed immune-checkpoint inhibitors, cisplatin-based chemotherapy remains the mainstay of treatment in both the neoadjuvant and metastatic setting. Chemotherapy portends a relatively modest 5–7% overall survival benefit in the neoadjuvant setting, and the overall response rate is well below 50% in metastatic [12,13]. These data taken together underscore the need to develop better treatments, and more importantly the need to predict individual patient therapeutic response.

Kidney cancer follows closely behind urothelial cancer as the eighth most common malignancy in the US. Again, like urothelial carcinoma, clinically localized stage I and II kidney cancers can be treated with extirpative management resulting in excellent survival. Unfortunately, treatment of locally advanced or systemic disease presents a significant challenge for clinicians, with a dismal 12% survival at 5 years in the metastatic setting [14]. As in urothelial carcinoma, immune-checkpoint blockade plays a growing role in treatment of advanced RCC. However, there are numerous targeted therapies that are currently employed in clinical practice or in development that require further investigation to delineate their role in treatment regimens. PDX models have the potential to define treatment modality and sequencing in both the localized and advanced settings for kidney and bladder cancers.

## 2. Experimental Procedures in PDX Engraftment

### 2.1. Host Animal Selection

Selection of immunocompromised mice for tumor engraftment plays an important role in the establishment of the PDX model. Researchers have developed successful models in four different types of mouse hosts: nude mice, scid mice, NOD (non-obese diabetic) scid mice, and mouse strains bearing targeted immunocompromising mutations in *Il2rg*.

Nude mice are one of the earliest mouse strains established for human cancer research, thus named for their lack of fur. They are athymic, and their reduction in T cell production inhibits the adaptive immune response. However, their intact innate immunity can limit utility for human tumor grafting, particularly in less aggressive, slower growing tumor types, or in more immunogenic malignancies such as RCC and urothelial carcinoma [15]. The scid mice are a strain with severe combined immunodeficiency developed in C.B17 mice, which prevents the development of mature T cells and B cells. This contributed to improved engraftment efficiencies over nude mice [16]. While nude mice and scid mice both lack adaptive immunity, their intact innate immunity and active natural killer (NK) cells can contribute to rejection of xenograft implantation and prevent tumor growth.

These issues with engraftment take-rate led to the development of more severely immunocompromised mice with impaired innate immunity, including NOD scid mice [17]. The lack of both innate and humoral immunity in these strains yields higher take-rates than nude or standard scid mouse strains, but these deficiencies lead to a shortened lifespan due to development of thymic lymphomas by 8–9 months of age, particularly in female mice. As such, careful monitoring of the animals for signs of spontaneous lymphoma development, including panting, bulged eyes, and splenic enlargement, is essential as thymic lymphoma can influence experimental results [18].

Finally, targeted mutations in *Il2rg* led to the development of NSG (NOD.Cg-Prkdc^scid^/l2rg^tm1Wjl^), NOG (NODShi.Cg-Prkdc^scid^/l2rg^tm1Sug^), and NRG (NOD.Cg-Rag1^tm1Mom^/l2rg^tm1Wjl^) strains with targeted deficiencies in B cells, T cells, and NK cells. The deficiency in both the innate and adaptive immune systems allows for acceptably high take rates for human tissue engraftment [19]. Though initially thought to be resistant to the spontaneous thymic lymphomas that limit the lifespan of NOD scid mice, various labs have reported low rates of both thymic lymphomas and other non-human tumors following engraftment. Development of spontaneous thymic lymphomas has been reported in up to 15% of C.B17 scid mice and 67% of NOD-scid mice, while the incidence of spontaneous lymphoma is much lower in *Il2rg*-deficient mouse models (NSG/NOG/NRG), with reported rates of 0.7% in NSG and NOG mice [15,20,21,22].

### 2.2. Tumor Specimen Collection and Preservation

Samples of primary and metastatic tumors sites that are resected surgically or biopsied are used for grafting. For RCC, sufficient tumor tissue can be taken at the time of radical or partial nephrectomy. After pathologic review, regions of the tumor specimen with viable (non-necrotic) malignant cells are brought to the lab for implantation. Unreliable tissue yields from tumor core biopsies make this a poor alternative. For bladder cancer, it is possible to establish grafts using tumor samples from both transurethral biopsy/resection specimens, as well as from radical or partial cystectomy specimens. Again, pathological confirmation by a genitourinary pathologist is a necessity, as chemotherapeutic treatment response or cautery artifact can affect tumor engraftment. Once specimens are removed from the patient, they should be placed in Roswell Park Memorial Institute (RPMI) medium or phosphate-buffered saline (PBS) solution and placed directly on ice. It is essential to minimize the time interval between excision of the tumor from the patient and engraftment in a mouse host in order to maximize tumor viability. In the case of renal cell carcinoma, when orthotopic engraftment into renal subcapsular space is planned, tissue slice grafts cut to 300 uM can be effectively used. Early models using this technique demonstrated ease of engraftment, and metastatic potential of the tumors. This approach also allows for generation of larger first-generation cohorts in the setting of limited tumor tissue [23].

While PDX models are often maintained through serial passaging in mice, this can be costly and labor intensive. With limited resources, it is often necessary to freeze early generation xenografts for potential expansion at a later date. Cryopreservation with subsequent reanimation is an efficient and effective way to select tumors for PDX. Samples can be preserved at −80 °C in cryoprotectant 10% DMSO (dimethyl sulfoxide) media with 10% fetal bovine serum and RPMI solution. Alternatively, preservation can be done in a specialized cryopreservative (i.e., Cryostor CS10–BioLife Solutions, Oswego, NY, USA), which has been shown to improve tumor reanimation engraftment efficiency [24,25]. Cryopreservation also gives researchers the opportunity to select tumors for PDX establishment or expansion based on clinical data that may not be known at the time of tissue acquisition.

### 2.3. Heterotopic Subcutaneous Engraftment

Once tumor tissue is adequately prepared for engraftment, host mice should be anesthetized using inhaled isoflurane. The area of fur overlying the anticipated incision should be shaved and sterilized using proper technique of betadine and alcohol. A small (<1 cm) skin incision is made in the flank or midback of the animal (Figure 1). A small subcutaneous space is developed bluntly using a surgical instrument. Care is taken not to damage the muscle. A single piece of tumor between 10–20 mm^3^ should be implanted into this incision; bilateral engraftment can be achieved through the same incision if desired. Bilateral engraftment can be a more cost-effective way to increase tumor yield. Wound clips or surgical suture should be used to close the incision once graft is placed. Institutional standards should be used to monitor the mice after surgical implantation. Caliper measurement devices are used to measure tumor size in 3 dimensions at least bi-weekly, though more frequent measuring is necessary if tumor growth is rapid. Peri-procedural collaborative involvement of the institutional veterinary staff is essential to ensure appropriate care and optimal results. When indicated by the veterinarians, mice may be started on prophylactic amoxicillin or trimethoprim-sulfamethoxazole in their food, or enrofloxacin (Baytril) in drinking water to prevent graft or surgical site infection. In such cases, the antibiotic is typically continued until the host animal is euthanized.

### 2.4. Orthotopic Engraftment

It has been suggested that orthotopic grafting, namely implantation of tumor tissue into the renal subcapsular space for RCC or into the bladder wall or lumen for urothelial cancer, may better simulate the clinical disease course with development of distant metastases [26]. While more challenging technically than subcutaneous implantation, renal subcapsular implantation takes advantage of the vascularity of the kidney and has been associated with higher engraftment rates, while potentially allowing for clinically relevant metastatic models to be created [18]. The approach involves a 5–7 mm incision parallel to and below the 13th rib. After dissecting through subcutaneous tissue and muscle, the kidney is pushed out through the incision [27]. Tumor cells can be put into a solution with Matrigel and injected into the renal subcapsular space using a small Hamilton needle, or a small incision in the capsule can be made and the donor tumor fragment can be placed under the capsule sheath.

Technical challenges specific to the murine bladder have limited the development of orthotopic PDX models, with subcutaneous heterotopic implantation in the midback or flank reported in most published studies to date. However, various methods of orthotopic bladder PDX model creation have been described in the literature. One such method utilizes a small laparotomy incision to exteriorize the bladder, followed by drainage and direct injection of tumor cells into the bladder lumen [28]. Alternatively, transurethral injection of cells into the bladder lumen has been described following an electrical or chemical perturbation of the bladder lumen, though inconsistent take-rates have largely limited this approach to patient-derived cell lines rather than bulk tumor implantation [29,30].

### 2.5. Advanced/Metastatic Disease Models

While evaluation of local tumor characteristics is important to guide therapeutic innovation, most patients eventually succumb to RCC and urothelial cancer due to metastatic spread. One of the major criticisms of PDX models, particularly heterotopic xenografts with subcutaneous implantation, is the lack of spontaneous metastases. Few metastatic models have been reported in bladder, though Tatum et al. described using BL0293, a patient-derived muscle-invasive bladder cancer PDX developed by Jackson Laboratory and University of California-Davis, as a reliable model for spontaneous and post-excisional metastatic disease to the liver. Notably, this model metastasized using a subcutaneous heterotopic implantation method [31]. In their cohort of orthotopic RCC xenografts, Thong et al. identified metastases to a few clinically relevant sites, including lung, liver, and bone, mimicking the clinical course of the patients from whom the tumors were derived [23].

A notable challenge in establishing a metastatic PDX model is non-invasive tracking of tumor spread. While cell-line derived xenografts can incorporate molecular markers such as luciferase to monitor for metastatic spread, this cannot be done with patient-derived tumor xenografts. Metastases can be identified on necropsy, but various imaging modalities allow for surveillance in survival studies. The primary tumor is monitored with calipers in the subcutaneous model, and orthotopic models can utilize ultrasound of the bladder or kidney, though image acquisition and interpretation are largely operator dependent. In their metastatic tumor model, Tatum et al. found that non-contrast T2-weighted MRI was the most effective way to accurately characterize metastatic burden in their mice [31].

### 2.6. Success Rates of Engraftment

Generally, successful engraftment rates depend on a combination of host animal selection and tumor histology. High grade and advance-stage tumors are typically associated with an improved take-rate, owing to aggressive characteristics of the tumor cells. The same characteristics that allow a tumor to seed distant metastatic sites in humans appear to also allow for successful engraftment in mouse hosts. The faster growth rate of aggressive tumors also allows for more rapid expansion and serial passaging [19]. In a study of 94 orthotopic RCC tumor graft samples, Sivavand et al. found an engraftment success rate of 80% when samples are derived from metastases, as opposed to a 14% take-rate when grafting from the primary tumor [32]. Their work suggested that a stable engraftment of a clinically localized tumor is correlated with metastatic potential and a shorter survival in the donor patient [32]. Slower growing tumors can be a challenge in establishment of PDX models. Use of Matrigel with growth factor for low-proliferation tumors may improve rate, though this remains a significant challenge.

Engraftment rates in renal cell carcinoma can vary widely depending on the site of engraftment. Orthotopic renal subcapsular engraftment of tumor into the murine kidney has been shown to recapitulate tumor growth characteristics and metastatic potential [23,33], but importantly this approach has also been shown to yield higher rates of successful tumor engraftment [19].

When the subcutaneous heterotopic approach is used to establish urothelial tumor PDX models, reasonably high take-rates can be expected. In a recent systematic review, Bernardo et al. looked at 12 studies using PDX models for urothelial carcinoma with an overall take rate of 41% [34], which compares favorably to those of generating PDX of prostate cancer or renal cell carcinoma [6,35]. The authors found that Matrigel, a commercially available cell culture matrix consisting of growth factors and extracellular matrix proteins, was also associated with an improved take-rate in bladder tumors. The Matrigel can also serve to prevent the grafted specimen from shifting within the subcutaneous pocket. While some researchers have attempted to utilize mechanical or enzymatic dissociation of the tumor cells followed by subcutaneous injection of a cell suspension, this has not been associated with improved engraftment rates when compared to solid tumor implantation [34].

To date, only six attempts at PDX creation from upper tract urothelial carcinoma (UTUC) have been reported with limited establishment and characterization [36]. As the genomic landscape of urothelial carcinoma is further elucidated, key differences have been identified in tumors of upper tract origin, differentiating them from lower urinary tract bladder cancers with potential biological and clinical implications [37,38]. Further work is needed to develop such PDX models.

## 3. In Vivo Therapy Response

### 3.1. Comparable Histology

Xenograft models can only serve as a translational and preclinical platform to accurately predict tumor response to therapies if they preserve histology, gene expression, and tumor mutation profile [18]. While the initial patient tumor histology should be confirmed with a genitourinary pathologist at the time of first engraftment, it is also essential to confirm maintenance of original histological characteristics with each successive passaging of tumor tissue. A small sample of representative tissue should be preserved and carefully examined to confirm malignant histology prior to experimental initiation. Numerous different histologic sub-types of RCC have been successfully grafted from human tumor samples, including conventional clear cell, papillary, translocation, and RCC with sarcomatoid differentiation [19]. Light microscopy evaluation of hematoxylin and eosin (H&E) stained slides, along with use of immunohistochemical staining can ensure that grafted tumors retain the same pathologic characteristics as the primary tumors. Immunophenotyping with molecular markers expressed by RCC, including carbonic anhydrase IX and CD10 have shown similar expression patterns, indicating successful concordance [33]. Importantly, in orthotopic RCC PDX models, malignant lymphocytes can infiltrate the kidney, mimicking growth of the tumor graft. This underlines the importance of verifying histology with every passage [18].

Following initial engraftment of the tumor into the host, human stroma is present with the malignant tumor cells, but with serial passages this is replaced by murine-derived stroma. When the endothelial cells, fibroblast, and other stromal elements known to affect cancer progression or inhibition are murine, there may be aberrant cross-reactivity with human-derived tumor cells. This has an impact on interactions with the tumor microenvironment, including alteration of paracrine signaling or murine cytokine reactivity. This limits the ability to examine therapies that target the tumor microenvironment in the PDX models and complicates the interpretation of drug response [39,40].

### 3.2. Retained Molecular Characteristics

It is essential that the PDX model recapitulates not only the phenotypic or histological characteristics of the in-situ disease, but that the genomic and transcriptomic alterations are maintained in the PDX model after engraftment and multiple passages. Testing of the somatic mutational profile of the PDX for concordance with the original tissue can ensure the fidelity of the engrafted tumor and its response to targeted therapies. While earlier studies sought to confirm the preservation of tumor identity with immunohistochemical staining, developments in next-generation sequencing (NGS) have lowered costs and allow researchers to evaluate passaged tumors to ensure concordance with parental tumor identity. More rapid testing with NGS yield results within weeks. Pan et al. demonstrated through whole-exome and transcriptome sequencing that their PDX model of bladder cancer shared 92–97% of the genetic aberrations of the parental tumor. This allowed them to identify druggable targets, such as FGFR3, to serve as targets of molecularly guided therapies. Such high fidelity, with preservation of the parental tumor genetic aberrations through serial passages, is extremely important to ensure translational applicability of observed treatment responses [41].

It has been shown that with serial passages, patient-derived cancer cell-lines will acquire DNA copy-number alterations that deviate from the parental tumor. This has been seen in RCC models as well [42,43]. Similarly, it is known that PDX models have the ability to acquire additional genomic mutations with subsequent passages. This has been observed across a variety of cancer types, including urothelial and renal cell carcinomas. PDX tumors can show divergent copy-number alteration (CNA) profiles from the parental tumor, suggesting different selection pressures altering the genomic profile of the PDX tumors. The potential impact of genetic drift should be considered with experimental design [44]. Retaining parental mutational profiles is particularly important with the development of therapies targeting patient tumors that harbor specific driver mutations. When pre-clinical validation of efficacy is dependent on genomic stability of the PDX model with serial passaging, it is prudent to systematically confirm the fidelity of the molecular profile.

Molecular subtyping of both bladder cancer and RCC is an area of clinical research with important implications for PDX models. Transcriptional clustering by mRNA expression analysis in bladder cancer can be used to categorize tumors into subtypes with distinct phenotypes. Within these subtypes are luminal, basal/squamous, and neuronal-type tumors. While further validation is needed, there are potential predictive and prognostic implications of molecular subtyping that may eventually guide therapy decisions. As such, it is important that the different subtypes be reflected in PDX models, as there may be clinically significant differences in sensitivity to chemotherapy or targeted therapies. For example, luminal cancers that tend to be enriched in activating mutations in FGFR3 may better respond to FGFR3 inhibiting agents [45].

Analogous to the molecular subtyping in bladder cancer, RCC expression profiles have been evaluated and may have similarly prognostic value. It is similarly imperative that RCC PDX models address differences in treatment responses seen in distinct subgroups identified using transcriptomic profiling [46]. Different molecular subtypes have been extensively correlated with treatment responses to targeted therapies, such as tyrosine kinase inhibitors [47]. Wang et al. used RNA-sequencing data from 35 patient-derived xenograft models to establish an “inflamed pan-RCC subtype.” Using a novel dissection algorithm that allowed them to define a tumor microenvironment gene expression signature, their work suggested the putative role of tumor cells in driving the stromal immune response [48].

An important limitation of genomic and transcriptomic sequencing in PDX models is the potential mixing of human and murine genetic content. To correct for this, reads mapping to human genes can be separated from reads mapping to mouse genes.

### 3.3. Correlation with Therapy Response and Chemosensitivity

The well-known limitations of human cancer cell lines have led researchers to search for preclinical cancer models that can reliably predict drug response in human trials. Alternatives to PDX include use of cell-line-derived xenografts, but these are known to lack heterogeneity of the disease in vivo. As discussed, they may acquire additional genetic alterations that lead phenotypic changes not reflected in the parental tumor [19]. PDX models might serve as a more consistent and accurate model of tumor response to treatment, primarily as a result of preserved tumor genomic profile and a more faithful model of intra-tumoral cellular heterogeneity [49]. In translational applications, panels of PDX models can screen for drug efficacy, resistance mechanisms, and aid in identifying biomarkers of predictive of drug sensitivity and response (Table 1). In a large cohort of mice grafted from a heterogeneous population of patient RCC tumors, Schueler et al. found that they were able to predict tumor response to therapy based on HMGB1 levels [50].

Traditional systemic chemotherapies have been notoriously ineffective against advanced renal cell carcinoma. As such, targeted therapies in RCC have become essential in the management of the disease when it has spread beyond the clinically localized setting. Numerous pathways including VEGF_1–3_, mTOR, PDGFRα, MET, FGFR_1–4_, RET, KIT, and AXL have been targeted by established and emerging therapies [12]. Various agents have been tested in the pre-clinical setting using PDX models to investigate tumor response. Importantly, it has been shown that PDX models for RCC can better correlate with clinical response to sunitinib than cancer cell lines [57]. In one of the largest studies to date, Sivavand et al. performed validation of targeted therapies in 94 different RCC patient tumor samples, 35 of which yielded viable tumor xenografts, and 16 were able to be serially passed successfully. Both histologic and genomic features were preserved in the xenograft models, allowing for testing of sensitivity to different targeted systemic therapies including sunitinib, sirolimus, and erlotinib [32].

In the treatment of urothelial cancers, platinum-based neoadjuvant chemotherapy (NAC) followed by radical cystectomy is the standard of care for muscle-invasive bladder cancer. While NAC has been associated with a pathologic response in 40% of patients with a complete response in up to 25%, there is significant overtreatment of non-responders [69]. The exposure to toxic agents without apparent benefit presents an opportunity to improve patient selection. Russell et al. demonstrated in their PDX model the heterogeneity of tumor response to these therapies, recapitulating the varied resistance seen in clinical practice [70]. With the recent addition of pembrolizumab, atezolizumab and other immune checkpoint inhibitors to the armamentarium of anti-neoplastic agents for urothelial carcinoma, there is a potential role for PDX models to serve as “mouse avatars,” allowing clinicians to determine platinum sensitivity and thus guide initial therapy choice in both the neoadjuvant and advanced settings. A number of published studies have sought to evaluate anti-neoplastic agents in urothelial carcinoma (Table 2).

At this time, approved therapy options targeting specific genes or biomarkers are limited in urothelial cancer. Erdafitinib, a tyrosine kinase inhibitor of fibroblast growth factor receptor (FGFR), was recently approved as the first targeted systemic therapy for metastatic urothelial cancer in patients with known FGFR2/3 mutations [71]. While no xenograft models have evaluated this drug, Pan et al. evaluated infigratinib (BGJ398), a similar selective pan-FGFR antagonist using bladder cancer PDX models. In addition, they evaluated other targeted treatments including EGFR/HER2 dual inhibitor lapatinib and PIK3CA inhibitor BEZ235, finding variable efficacy [41]. Jager et al. similarly evaluated FGFR3 inhibition using a monoclonal antibody in PDX models [72].

### 3.4. Future Directions

In the future, select patients with high-risk, clinically localized disease may be potential candidates for personalized PDX models. Tumor samples can be collected at the time of radical cystectomy (for urothelial carcinoma) or nephrectomy (for RCC) to establish xenograft models. The PDX model can then be used to screen both standard therapies and drugs targeting specific molecular aberrations in the patient’s tumor. If the patient presents with disease recurrence, which typically takes several months, drugs with proven efficacy in the PDX models can be administered to the patient. Further studies demonstrating correlation between PDX response and patient parental tumor response are needed to validate such an approach. Furthermore, the cost of such as system has thus far prevented such integration into clinical practice. A more realistic approach is to generate PDX libraries with representative tumor subtypes and mutational variants. A focus on models that have been under-represented in cell lines would better reflecting the intertumoral heterogeneity of RCC and urothelial cancer. Once a library is established, specific targeted therapies can be screened for efficacy and resistance mechanisms.

## 4. Limitations

An understanding of the limitations of PDX models is essential for appropriate experimental design. When compared to traditional cancer cell lines, PDX models are costly and require substantial technical expertise. They can also be time consuming, with slow-growing tumors requiring months for successful engraftment.

### 4.1. Immunotherapy Testing

Immuno-oncology is an exciting and rapidly evolving field, particularly with the treatment of advanced bladder and kidney cancers. This therapeutic revolution has seen the recent approval of numerous immune checkpoint inhibitors [12]. Unfortunately, the utilization of immune-deficient mice limits the ability to test therapies that rely on T cell anti-tumor activity, significantly restricting the applicability of PDX models in the testing of immune checkpoint inhibitors. The development of humanized mice has sought to resolve this significant disadvantage. These mice, also known as human haemato-lymphoid chimeric mice or human immune system models, generate a competent human immune system through various methodologies [40]. This has been an important development in the field, accomplished by irradiating NSG or NOG mice and then transplanting human CD34+ hematopoietic progenitor and stem cells. The mice are considered humanized when over 25% of the cells in the peripheral blood are hCD34+. These humanized NSG mice can develop a partially functional human immune system, allowing for better characterization of the tumor-immune microenvironment and the testing of immunotherapy drugs. Limited studies have evaluated the efficacy of immunotherapies in bladder cancer cell lines. Wang et al. showed that by using allogeneic, but HLA partially matched CD34+ hematopoietic stem cell donors and tumors (including bladder cancer), immune checkpoint inhibitors can be effectively studied in the humanized mouse PDX model. In their study, the mice did not reject the engrafted tumors and demonstrated the efficacy of pembrolizumab in CD8+ T-cell mediated tumor growth inhibition [81]. Blinova et al. used humanized NSG mice to test PDL-1 blocker durvalumab and demonstrated PDL-1 expression in PDX models with significant stagnation of tumor growth [73]. It is hoped that further development of humanized mouse PDX models will allow for the study of the interaction of the tumor and the human immune system, as early models suggest that both therapeutic outcomes and side effects can be recapitulated [82].

### 4.2. Xenotropic Murine-Derived Viral Infection

It is important for any researcher working with patient-derived xenografts or patient-derived cell lines to be aware of the potentially confounding effect that xenotropic murine retroviruses (XMRV) can have on growth parameters. There are reports of xenotropic murine leukemia viruses (MLVs) that are activated in immunodeficient mice with passage of human tissue. Though initially thought to play a causal role in the oncogenesis of prostate cancer, this has since been debunked. However, xenotropic viruses such as MLV, when activated, can induce spontaneous tumor formation in mice. It is suspected that there is relatively high prevalence of xenotropic viral infection in PDX models, with one study reporting detection of xenotropic endogenous murine leukemia virus in up to 53% of human breast cancer PDXs. Though the effect of XMRVs specifically on growth of kidney and urothelial tumors is unknown, PCR should be used to detect activated virus if there is any clinical suspicion. Periodic monitoring for XMRVs can also be done if there is concern for contamination. This also emphasizes the importance of sterile technique with tumor passaging to prevent widespread viral contamination of PDX models or cell lines in the lab [83,84,85].

### 4.3. Intratumoral Heterogeneity

The intertumoral heterogeneity that is seen in clinical practice is not reflected in genetically engineered mouse models, which tend to develop more homogeneous tumors. Tumor grafts are derived from a small area of the parental tumor, and thus fail to fully capture the heterogeneity of the entire tumor. Sampling multiple areas of the tumor for grafting can better capture the intra-tumoral heterogeneity that is known to be present in both bladder cancer and RCC, but this remains a limitation of any pre-clinical model.

## 5. Alternative Patient-Derived Models: Organoids

Building on experience with patient-derived cell lines, the utilization of patient-derived organoids (PDO) provides a less expensive alternative to solid tumor xenograft models. Organoids are three-dimensional tissue cultures and can be successfully established using human tumor cells. Lee et al. described 22 patient-derived bladder cancer organoid lines, and they successfully converted these lines into orthotopic xenografts using ultrasound-guided implantation. They demonstrated the ability to efficiently interconvert organoids and xenograft models in urothelial carcinoma [86]. Furthermore, they found high concordance of molecular profiles in the organoid and xenograft models with the parental tumors and were able to validate drug responses in vivo. As a result of the identified interconvertibility, they proposed a model of disease treatment in which patients have drug responses identified in PDOs, which can then be validated in PDX models prior to guiding patient therapies. Similarly, Mullenders et al. established a biobank of patient-derived urothelial tumor organoids that showed basal and luminal cells and were able to use them as predictors of chemotherapy response. Their high efficiency of human bladder cancer organoid generation allowed for extensive experimentation and the potential for novel drug screening [87]. Clearly, there will be a continued important role for patient-derived organoids to be used in conjunction with patient-derived xenografts in development of targeted therapies.

## 6. Conclusions

While no single in vivo system fully recapitulates the complexity of human cancers, combined pre-clinical approaches that include patient-derived xenografts are a growing and important component for research in urothelial and renal cell carcinomas. With the expanding number of both therapeutic drugs and better characterization of rational targets in urologic malignancies, the role for PDX models in elucidating susceptibilities and mechanisms of treatment resistance is expanding. Advancements in technical efficiency, improved understanding of limitations, and clinical validation of PDX models in bladder and renal malignancies will allow for improved translational application.

## Figures and Tables

**Figure 1 cancers-12-00439-f001:**
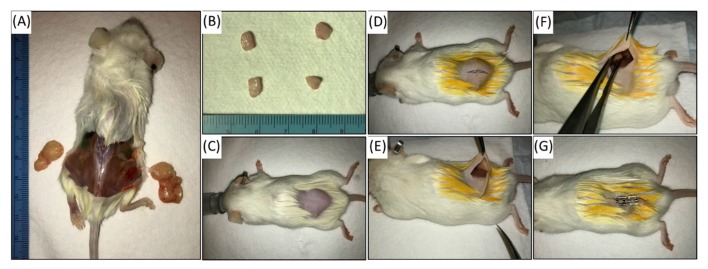
Passaging of patient-derived xenograft (PDX): NSG mice are euthanized once tumors reach approximately 1.5cm as measured by Vernier caliper (**A**). Tumors are explanted and divided into 5mm pieces for engraftment (**B**). Additional tumor specimen is collected at every passage for formalin fixation and paraffin embedding to confirm faithful maintenance of malignant histology. The new host NSG mouse at 6-8 weeks is anesthetized and prepped for grafting (**C**). The hind region is shaved and cleaned with alcohol and betadine. A 1cm midline incision allows for bilateral implantation, if desired (**D**,**E**). Tumor specimen is placed in subcutaneous pouch overlying the muscle (**F**). Matrigel can be used to keep tumor in position. Skin is closed, and mouse is started on amoxicillin for infection prophylaxis (**G**).

**Table 1 cancers-12-00439-t001:** Published studies of targeted anti-neoplastic treatments utilizing novel patient-derived tumor xenografts models in renal cell carcinoma.

Study Name	Cancer Type	PDX Tumor Models	Graft Location	PDX Line	Targets	Drugs/Therapies
Elbanna et al. (2019) [51]	Clear cell renal cell carcinoma	3	Orthotopic and subcutaneous heterotopic	RP-R-01, RP-R-02, and RP-R02LM	Angiopoietin 1/2, MET kinase	Trebananib (angiopoietin 1/2 inhibitor), MET kinase inhibitor
Schueler et al. (2018) [50]	Clear cell, papillary, chromophobe renal cell carcinoma	44	Subcutaneous heterotopic	Institutional: University Hospital Frankfurt	VEGF, VHL-associated targets, mTOR	Sunitinib, pazopanib, sorafenib, axitinib, temsirolimus, bevacizumab
Adelaiye-Ogala et al. (2018) [52]	Clear cell renal cell carcinoma	2	Subcutaneous heterotopic	RP-R-02LM, 786-O	Androgen receptor, receptor tyrosine kinase	Enzalutamide, sunitinib
Damayanti et al. (2018) [53]	Translocation renal cell carcinoma	1	Subcutaneous heterotopic	RP-R07	PI3K/AKT/mTOR pathways	Rapamycin, MLN0128 (mTOR inhibitor), BEZ-235 (PI3K inhibitor)
Zhao et al. (2017) [54]	Papillary renal cell carcinoma	1	Orthotopic and subcutaneous heterotopic	Institutional tumor	MET	Cabozantinib
Adelaiye-Ogala et al. (2017) [55]	Clear cell renal cell carcinoma	2	Ectopic in prostate (metastatic model), Orthotopic, and Subcutaneous heterotopic	RP-R-01, RP-R-02, and RP-R02LM	EZH2, VEGF	HKT288, sunitinib, axitinib, bevacizumab
Bialucha et al. (2017) [56]	Clear cell renal cell Carcinoma	3	Subcutaneous heterotopic	Multiple institutional tumors, commercial vendors	CDH6	HKT288 (anti-CDH6 antibody drug conjugate)
Dong et al. (2017) [57]	Renal cell Carcinoma	33	Subcutaneous heterotopic	Institutional: Memorial Sloan Kettering Cancer Center (New York)	Receptor tyrosine kinase	Sunitinib
Hong et al. (2017) [58]	Renal cell Carcinoma	2	Subcutaneous heterotopic	Institutional: Peking University Hospital (Peking)	PDGFA, PDGFB, PDGFRA	Sorafenib, sunitinib, axitinib
Chen et al. (2016) [59]	Renal cell carcinoma	22	Orthotopic	Institutional: UT Southwestern (Dallas, TX)	HIF-2	PT2399 (HIF-2 antagonist), sunitinib
Diaz-Montero et al. (2016) [60]	Renal cell carcinoma	2	Subcutaneous heterotopic	Institutional: Cleveland Clinic (Cleveland, OH)	MEK1/2	Sunitinib, PD-0325901 (MEK inhibitor)
Lang et al. (2016) [61]	Renal cell carcinoma	30	Orthotopic and Subcutaneous heterotopic	Institutional: Hôpitaux Universitaires de Strasbourg (France)	VHL-associated targets	Sunitinib, sorafenib, everolimus
Adelaiye et al. (2015) [62]	Clear cell renal cell carcinoma	2	Subcutaneous heterotopic	RP-R-01 and RP-R-02	Receptor tyrosine kinase	Sunitinib
Ciamporcero et al. (2015) [63]	Renal cell carcinoma	1	Subcutaneous heterotopic	RP-R-01	VEGF and HGF/c-met pathway	Axitinib, crizotinib, sunitinib
Schuller et al. (2015) [64]	Papillary renal cell carcinoma	2	Subcutaneous heterotopic	RCC-43b and RCC-47 PRCC	MET	Savolitinib, sunitinib
Miles et al. (2014) [65]	Clear cell renal cell carcinoma	2	Subcutaneous heterotopic	RP-R-01 and RP-R-02	DII4, VEGF,	REGN (mAb binding DII4), ziv-aflibercept (VEGF blocker), sunitinib
Thong et al. (2014) [23]	Renal cell carcinoma	13	Orthotopic	Institutional: Stanford Hospital (Stanford, CA)	Receptor tyrosine kinase	Sunitinib
Ingels et al. (2014) [66]	Renal cell carcinoma	3	Orthotopic	Institutional: Stanford (Stanford, CA)	mTOR	MLN0128 (mTOR inhibitor), temsirolimus
Sivanand et al. (2012) [32]	Renal Cell Carcinoma	35	Orthotopic	Institutional: UT Southwestern (Dallas, TX)	VHL-associated targets	Dovitinib, sirolimus, sunitinib
Karam et al. (2011) [67]	Renal cell carcinoma	4	Orthotopic and Subcutaneous heterotopic	Institutional: MD Anderson Cancer Center (Houston, TX)	VHL-associated targets	Sunitinib, everolimus
Hammers et al. (2010) [68]	Renal cell carcinoma	1	Subcutaneous heterotopic	Institutional: Johns Hopkins (Baltimore, MD)	Receptor tyrosine kinase	Sunitinib

**Table 2 cancers-12-00439-t002:** Published studies of targeted anti-neoplastic treatments utilizing novel patient-derived tumor xenografts models in urothelial cell carcinoma.

Study Name	Cancer Type	PDX Tumor Models	Graft Location	PDX Line	Target	Drug/Therapy
Blinova et al. (2019) [73]	Urothelial cell carcinoma	6	Subcutaneous heterotopic	Institutional: National Research Medical Center of Radiology (Moscow)	PD-L1	Durvalumab
Zeng et al. (2017) [74]	Urothelial cell carcinoma	3	Subcutaneous heterotopic	BL0269, BL0293, BL0440 (UC Davis/Jackson Labs)	PI3K pathway	Pictilisib, Cisplatin, gemcitabine
Ler et al. (2017) [75]	Urothelial cell carcinoma	Not reported	Subcutaneous heterotopic	Institutional: Singapore General Hospital (Singapore) and Chang Gung Memorial Hospital (Taiwan)	EZH2	GSK503 (EZH2 methyltransferase inhibitor)
Wei et al. (2016) [76]	Urothelial cell carcinoma	2	Subcutaneous heterotopic	Institutional: Roswell Park BLCAb001, BLCAb002	PI3K/mTOR	Cisplatin, LY414 (dual PI3K/mTOR inhibitor)
Chang et al. (2016) [77]	Urothelial cell carcinoma	1	Subcutaneous heterotopic	Institutional: Samsung Medical Center (Seoul)	SRC and PI3K/AKT/mTOR	Dasatinib, PKI-587 (dual PI3K/mTOR inhibitor)
Pan et al. (2016) [78]	Urothelial cell carcinoma	1	Subcutaneous heterotopic	Institutional (UC Davis)	Bladder cancer cells (PLZ4 ligand)	Disulfide-crosslinked PLZ4-nanomicelle paclitaxel
Ciamporcero et al. (2016) [79]	Urothelial cell carcinoma	2	Subcutaneous heterotopic	Institutional: Roswell Park BLCAb001, BLCAb002	YAP (Yes-associated protein)	Verteporfin, cisplatin
Guo et al. (2016) [80]	Urothelial cell carcinoma	1	Subcutaneous heterotopic	Institutional: Shanghai Changhai Hospital (Shanghai)	HuR RNA-binding protein	Pyrvinium pamoate combined with cisplatin
Jager et al. (2015) [72]	Urothelial cell carcinoma	7	Renal subcapsular	Institutional: Vancouver General Hospital (Vancouver, Canada)	FGFR3	R3Mab (anti-FGFR3 antibody)
Pan et al. (2015) [41]	Urothelial cell carcinoma	22	Orthotopic and Subcutaneous heterotopic	Institutional: UC Davis (including PDX-BL0293, PDX-BL0382)	EGFR/HER2, PIK3CA, FGFR3	Lapatinib, ponatinib, BEZ235 (PI3K/mTOR inhibitor), BGJ398 (FGFR inhibitor)
Cirone et al. (2014) [3]	Urothelial cell carcinoma	2	Subcutaneous heterotopic	Commercial: PDX-BL0293, PDX-BL0382 (UC Davis/Jackson Labs)	PI3K/mTOR, MEK	PF-502 (PI3K/mTOR inhibitor), PD-901 (MEK inhibitor)
Abe et al. (2006) [36]	Urothelial cell carcinoma	15	Subcutaneous heterotopic	Institutional: Hokkaido University Hospital	n/a	Radiation

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
