# Peer review of "Patient-Derived Xenograft Models in Urological Malignancies: Urothelial Cell Carcinoma and Renal Cell Carcinoma"

_cancers, 2020, doi:10.3390/cancers12020439_

Round 1

Reviewer 1 Report

the authors addressed thoroughly all comments and improved the quality of the paper significantly

Author Response

Thank you for the feedback and review.

Reviewer 2 Report

The modifications made make for a stronger paper.

Author Response

Thank you for the feedback.

Reviewer 3 Report

Dear authors,

The manuscript “Patient-Derived Xenograft Models in Urological Malignancies: Urothelial Cell Carcinoma and Renal Cell Carcinoma” was extensively re-written. Most of the limitations pointed in the first manuscript have been addressed in the current version, which has been substantially improved.

Minor aspects to be corrected:

Line 137: … costly “and” labor intensive…

Line 443: …predictors “of” chemotherapy…

Line 249: “tumor mutation profile” instead of “DNA copy-number alterations” (there are other DNA changes that can occur and are relevant as described later in the manuscript)

Line 253: consider deleting the part “on a slide with successive passages” to improve clarity.

Lines 297: please change sentences to clarify their meaning. “continual identification of fidelity of genomics”

The tables are not cited in the text.

Citation error: [TATME] several times along the text

Best regards,

Author Response

Minor aspects to be corrected:

Line 137: … costly “and” labor intensive…

Line 443: …predictors “of” chemotherapy…

Line 249: “tumor mutation profile” instead of “DNA copy-number alterations” (there are other DNA changes that can occur and are relevant as described later in the manuscript)

Line 253: consider deleting the part “on a slide with successive passages” to improve clarity.

Lines 297: please change sentences to clarify their meaning. “continual identification of fidelity of genomics”

The tables are not cited in the text.

Citation error: [TATME] several times along the text

The above-listed errors and suggestions have been addressed and the new changes are reflected in the updated manuscript submission. Thank you for your comments and review.

Reviewer 4 Report

The authors responsed appropriately to the commnets of the reviewer's and I think acceptable for publication.

Author Response

Thank you for the feedback and comments.

This manuscript is a resubmission of an earlier submission. The following is a list of the peer review reports and author responses from that submission.

Round 1

Reviewer 1 Report

Abstract: why do we need humanized PDX for targeted therapy development and what is the link to personalized medicine in this context?

Line 45: please use the correct mouse strain nomenclature

Line 99: the authors should comment on spontaneous lymphomas in NSG/NOG/NRG backed up with literature

Line 114: “Studies have shown that tumor samples should be processed and used directly for implantation between 1-24 hours after collection, though shorter intervals for implantation are optimal.” Please clarify if this also covered by reference nr 16

Line 131: the treatment of the animals with antibiotics is not compliant with a number of animal welfare rules. The medication of the animal has to be decided by a vetenarian

The wording “implantation in the hindleg” should be avoided it is misleading. I assume that subcutaneous implantation is meant.

Legend Figure 1: in the legend the authors describe a 2 cm incision, in the text an incision with < 1 cm is described. The larger incision is not compliant with animal welfare rules specifically if small mice e.g. NSG are used.

Line 179: UTUC please provide complete abbreviation list and introduce those

Line 260: “While no xenograft models have evaluated this drug, Pan et al evaluated infigratinib (BGJ398), a similar selective pan-PFPR antagonist using bladder cancer PDX models.” The authors should comment on the fact that in PDX the stroma is of murine origine as it is in cell line derived xenografts

Line 264: check for typo in the abbreviation

Table 1: This reference is missing: https://www.ncbi.nlm.nih.gov/pubmed/30123419

General: to avoid discussion about animal welfare standards in Europe, US and worldwide the authors should avoid going too much into technical details. The exact procedures have to fit to the regulations in the respective country/institution

Reviewer 2 Report

Suggestions to improve the manuscript:

There are several places where abbreviations need to be defined prior to use: RCC, NSG/NOG/NRG, UTUC, PDO. Line 131-134, with proper aseptic technique it is not a routine practice to add antibiotics post-surgery, especially when treating mice with other cancer-targeting drugs. The authors should clarify that these approaches are only used sometimes. Line 134, switch “sacrificed” to “euthanized”. Line 161-184, or line 221-241, the authors should comment on the growth rate of PDXs grown in the orthotopic location as compared to subcutaneous, and further, how these differences correspond with drug effectiveness, if known. Along these lines, in Table 1, perhaps an asterisk, or some other notation, could be used to denote the organ/location that the PDX was grown; orthotopic or subcutaneous. Line 174, change to “take-rate”. Line 198/section 3.2, the authors should comment on how the mouse and human genetic content could potentially effect the results of NGS studies, and if any studies accounted for this in their analyses. Line 214, remove “model”. Line 221/section 3.3, it could be beneficial to describe common approaches used for drug administration; IP, IV, subcutaneous, in the food or water, etc. Do the studies listed in Table 1, mimic how drugs are administered in the clinic? Line 259, remove “been”. Line 289, add “of” between efficacy/immunotherapies. Line 290, place a comma after “allogeneic”. Since this review will be a good reference for these two cancer types, the authors should clarify if there were any overlapping PDXs used across the different studies.

Reviewer 3 Report

The manuscript entitled: Patient-Derived Xenograft Models in Urological Malignancies: Urothelial Cell Carcinoma and Renal Cell Carcinoma seeks to review the role of PDX models in urothelial and kidney cancer research and discuss methodologies, applications and future directions.

The subject of the manuscript is relevant and worthy however, there are several problematic issues with the way the evidence is presented and discussed.

PDX models are presented as a way to guide ´precision medicine´ as they better retain the complexity and molecular characteristics of the primary tumors opposed to the standard use of cancer cell lines. It would be relevant to discuss the details of the whole process and what would have to happen to actually make it a viable option is such setting. Currently, there are no reports of routine use of PDX models to guide therapeutic decision, not only for urothelial and kidney cancer but for any solid malignancy. This has been discussed in many of the original papers on the subject and also recently reviewed here:  https://doi.org/10.1016/j.euo.2018.08.014.  “The establishment of PDX models is a relatively lengthy and costly process, generally taking 4–6 mo, which is outside the clinical window for treatment intervention. It also requires a large number of donor cells and inoculated hosts to reflect patient-to patient genetic variability. In many clinical cases, only a limited amount of biopsy tissue is available, which reduces the chance of successful engraftment”.

The authors of the current manuscript highlight the potential of PDX models for “for real-time clinical decision making and treatment selection” in several parts of the manuscript. Here: “The development of PDX models are proving to be particularly useful in the application of precision medicine within urologic malignancies” and “This would require establishing tumor models in real-time, which 253 has been a challenge in the field, though improvements in technique”. A more detailed and critical discussion of the practicalities of such application would be essential to support the claims. Also, how to handle low proliferative tumors?

Overall the manuscript is vague, lacks detail and scientific discussion of the literature mentioned which significantly impairs the usefulness of the review. What would be impact of the different molecular subtypes of bladder and renal cancers in the ability to establish and use such models. What are the authors considerations about intertumoral heterogeneity. The potential and advantages of the use of PDX models for translational research is not discussed apart from their use for drug screening in the clinics which is highly problematic.

Statements that are not supported by current literature:

Line 45: “When utilized for research purposes, many studies have shown that these tumors in immune compromised mice retain the original tumor histology, cellular structure, tumor microenvironment, and genomic characteristics.” The tumor microenvironment in the PDX models is actually one of the aspects that is not preserved. Although it is a more complex and complete model when compared with in vitro model systems, the human stroma components of the original tumor are replaced by stroma cells of mouse origin after the first few passages and immune components are absent. This aspect needs to be corrected and further discussed in the manuscript.

Line 131: “Immediately following engraftment, mice are started on prophylactic amoxicillin or trimethoprim-sulfamethoxazole in their food, or enrofloxacin (Baytril) in drinking water to prevent graft or surgical site infection. The antibiotic is continued until the host animal is sacrificed.” When standard protocols are followed, the tumor pieces and all surgical procedures are under sterile conditions there is no need for antibiotic treatment. Furthermore, the use of antibiotics in this settings would further complicate the analysis of tumor response and increase the number of confounding factors as a result of drugs metabolism and interaction.

The description of the methods is vague and not accurate. Considering that there are several papers and detailed protocols published on this subject I would recommend focus on scientific discussion of critical aspects instead.

Line 151: “Alternatively, transurethral injection of cells into the bladder lumen has been described following an electrical or chemical perturbation of the bladder lumen, though inconsistent take-rates have largely limited this approach to patient-derived cell lines rather than bulk tumor implantation” The aim of orthotopic implantation is to establish the tumor growth in the primary location, i.e. bladder wall. To achieve that both bladder instillation with tumor cells and intramural inoculation of cells have been described in several papers. This part in the manuscript needs substantial re-writing.

Reviewer 4 Report

The authors summarized patient-derived xenograft models of urothelial cell carcinoma and renal cell carcinoma. Generally, the article is well-written and accessible to a wide audience.
However, I found the review superficial in some issues.

The section on 2.4 the authors described success rates of engraftment. However, we need more specifically take-rates with regards to both cancers.

One of the shortcomings not mentioned by the authors is the paucity of models in which the tumors metastasize. This is an enormous area of concern for clinicians.

Another shortcoming is the problem that xenotrophic mouse-borne viruses can infect the human tumor tissues, and this could result in differences in tumor behaviors which do not reflect what happens in the patient who donated the tissue.

The authors did not mention to technique for cryopreservation. This is very important issues to keep biological and genetical characteristics that the original tumors have.
